# Fabrication of a Novel CNT-COO^−^/Ag_3_PO_4_@AgIO_4_Composite with Enhanced Photocatalytic Activity under Natural Sunlight

**DOI:** 10.3390/molecules28041586

**Published:** 2023-02-07

**Authors:** Abdalla A. Elbashir, Mahgoub Ibrahim Shinger, Xoafang Ma, Xiaoquan Lu, Amel Y. Ahmed, Ahmed O. Alnajjar

**Affiliations:** 1Department of Chemistry, College of Science, King Faisal University, Al-Hofuf 31982, Saudi Arabia; 2Department of Chemistry, Faculty of Science, Khartoum University, P.O. Box 321, Khartoum 11111, Sudan; 3Department of Applied and Industrial Chemistry, Faculty of Pure and Applied Sciences, International University of Africa, Khartoum 11111, Sudan; 4Key Laboratory of Bioelectrochemistry & Environmental Analysis of Gansu Province, College of Chemistry & Chemical Engineering, Northwest Normal University, Lanzhou 730070, China; 5Chemistry and Nuclear Physics Institute, Sudan Atomic Energy Commission, P.O. Box 3001, Khartoum 11111, Sudan

**Keywords:** insitu synthesis, CNT-COO^−^, Ag_3_PO_4_@AgIO_4_, sunlight, organic dyes

## Abstract

In this study, a carboxylated carbon nanotube-grafted Ag_3_PO_4_@AgIO_4_ (CNT-COO^−^/Ag_3_PO_4_@AgIO_4_) composite was synthesized through an in situ electrostatic deposition method. The synthesized composite was characterized by Fourier transform infrared (FT-IR) spectroscopy, X-ray diffraction (XRD), scanning electron microscopy (SEM), diffuse reflectance spectroscopy (DRS), and energy-dispersive X-ray spectroscopy (EDS). The electron transfer ability of the synthesized composite was studied using electrochemical impedance spectroscopy (EIS). The CNT-COO^−^/Ag_3_PO_4_@AgIO_4_ composite exhibited higher activity than CNT/Ag_3_PO_4_@AgIO_4_, Ag_3_PO_4_@AgIO_4_, and bare Ag_3_PO_4_. The material characterization and the detailed study of the various parameters thataffect the photocatalytic reaction revealed that the enhanced catalytic activity is related to the good interfacial interaction between CNT-COO and Ag_3_PO_4_. The energy band structure analysis is further considered as a reason for multi-electron reaction enhancement. The results and discussion in this study provide important information for the use of the functionalized CNT-COOH in the field of photocatalysis. Moreover, providinga new way to functionalize CNT viadifferent functional groups may lead to further development in the field of photocatalysis. This work could provide a new way to use natural sunlight to facilitate the practical application of photocatalysts toenvironmental issues.

## 1. Introduction

Since the application of semiconductors as photocatalysts was discovered [1], enormous efforts have been madeto develop and enhance their catalytic performance by discovering new photosensitized material or designing heterogeneous semiconductor photocatalysts [2,3,4,5,6,7,8,9,10,11,12]. Recently, Ag_3_PO_4_ has been discovered as a novel semiconductor photocatalyst with exclusive photocatalytic activity toward oxygen evolution and organic pollutant removal [13,14,15]. However, its highest possibility of photocorrosion stands as the main drawback that hinders its practical application [13,16,17]. Therefore, it isnecessaryto overcome this drawback by designing and constructing Ag_3_PO_4_-based composites. Recently, AgIO_4_ withconduction and valence bands of 3.61 eV and 0.96 eV, respectively, has beenreported as a promising visible light-responsive photocatalyst [18]. Therefore, fabricating Ag_3_PO_4_ with AgIO_4_ can successfully improve charge separation and transportation.

Carbon-based substrates played an important role in supporting the photocatalytic performance of the photocatalysts due to their novel chemical, thermal, electrical, and optical properties. Carbon nanotubes (CNT), which are fabricated from sp2 carbon, have attracted researchers’ attention due to their unique specific surface area (100–700 m^2^ g^−1^), and electronic, optical, chemical, and thermal characteristics [12,19,20,21,22,23,24,25]. These properties recommend CNT as a novel material for catalyst carriers and supporters in heterogeneous catalysts. Some studies have reported improved photocatalytic activity for CNT-based composites [26,27,28,29,30,31]. However, the low dispersion and the poor interfacial interaction between CNT and other materials remain the major disadvantages that limit their practical application as catalyst supporters [32,33,34]. Therefore, functionalization of CNT achieved by attaching electronegative groups or aliphatic carbon chains may increase their dispersion. Furthermore, this process can provide a more active surface on the CNT, allowing for a greater amount of catalyst materials to be loaded onto the CNT’s surface.Based on the above analysis, CNT might be used as an electron capture agent to increase the activity and stability of Ag_3_PO_4_.Therefore, developing a high-efficiency photocatalyst is always a hot research topic in the photocatalytic field. In this study, CNT-COO^−^/Ag_3_PO_4_@AgIO_4_ was synthesized with enhanced photocatalytic activity through an in situ electrostatic deposition method, which was then characterized using different techniques. The photocatalytic activity of the synthesized CNT-COO/Ag_3_PO_4_@AgIO_4_ was evaluated based onthe degradation of methylene blue as the target dye under natural sunlight. In addition, the effect of the optimal CNT-COOH amount on the photocatalytic performance was also investigated. Moreover, the mechanism of photodegradation was proposed.

## 2. Results and Discussion

### 2.1. The Synthesis Mechanism of CNT-COO^−^/Ag_3_PO_4_@AgIO_4_

In this work, CNT-COO^−^/Ag_3_PO_4_@AgIO_4_ was synthesized viathe electrostatic deposition method, as shown diagrammatically in Figure 1. After the negatively charged CNT-COO was suspended in water and the Ag^+^ ions were added, the electrostatic interaction derived the adsorption of the positively charged Ag^+^ ions onto the negatively charged CNT-COO to form intermediate complexes of CNT-COO^−^/Ag^+^. After Na_2_HPO_4_ was dropped into the mixture, the HPO_4_^2−^ could react with Ag^+^ ions on the surface of CNT-COO to form Ag_3_PO_4_ nuclei. As the reaction proceeded, Ag_3_PO_4_ particles successfully grew on the surface of CNT-COO. The addition of KIO_4_, which was added dropwise, into the solution mixture resulted in the formation of large-sized AgIO_4_ particles, which were grafted by CNT-COO^−^/Ag_3_PO_4_ to form the CNT-COO^−^/Ag_3_PO_4_@AgIO_4_ composite.

### 2.2. Characterization

#### 2.2.1. FT-IR Analysis

The as-synthesized materials were first characterized by FT-IR, and the results areshown in Figure 2a. Non-carboxylated CNT does not have any characteristic peaks, whereas the carboxylated one (CNT-COOH) shows CO stretching vibration at about 1700 cm^−1^, and the ^−^OH groups show stretching and bending vibrations at about 3450 cm^−1^ and 1615 cm^−1^, respectively, confirming the successful carboxylation of CNT. In the FTIR spectra of AgIO_4,_ a band located between 626–830 cm^−1^ is attributed to the vibrations between Ag-I-O atoms. The ^−^OH group of the physically absorbed H_2_O shows stretching and bending vibrations at 3450 cm^−1^ and 1630 cm^−1^, respectively. In the spectra of Ag_3_PO_4_, the peak located at 553 cm^−1^ is attributed to the bending vibration of the O=P-O. The symmetric and asymmetric stretching vibrations of P-O-P rings are found at 853 and 998 cm^−1^, respectively. The absorbed H_2_O molecules are shown to have bending and stretching vibrations at 1662 and 3420 cm^−1^, respectively. The P=O shows stretching vibration at 1393 cm^−1^. In the spectra of CNT-COO^−^/Ag_3_PO_4_@AgIO_4_ and Ag_3_PO_4_@AgIO_4_, all the characteristic vibration peaks of the individual materials CNT-COO^−^, Ag_3_PO_4_, and AgIO_4_ are present, which proves the formation of the heterocomposite.

#### 2.2.2. XRD Analysis

The phase purity and the crystal structure of AgIO_4_, Ag_3_PO_4_, Ag_3_PO_4_@AgIO_4_, and CNT-COO^−^/Ag_3_PO4@AgIO_4_ composites are identified by XRD spectra, which are shown in Figure 2b. Pure Ag_3_PO_4_ shows characteristic diffraction peaks that are identical to the body-centered cubic structure of Ag_3_PO_4_ [35].The XRD pattern of AgIO_4_ diffraction peaks can be indexed to the tetragonal structure of AgIO_4_ [36]. TheXRD results indicate that the well-crystallized Ag_3_PO_4_ and AgIO_4_ were successfully fabricated under experimental conditions. For Ag_3_PO_4_@AgIO_4_, theXRD patterns showed diffraction peaks corresponding to Ag_3_PO4 and AgIO_4_ crystal phases, proving that AgIO_4_ and Ag_3_PO_4_ are well coupled and that Ag_3_PO_4_@AgIO_3_ has been successfully synthesized. In the pattern of CNT-COO^−^/Ag_3_PO_4_@AgIO_4_, a smalldecrease is observed compared with Ag_3_PO_4_@AgIO_4_, suggesting that the combination of CNT-COOH does not affect the crystalline structure and phase composition of Ag_3_PO_4_@AgIO_4_; therefore, no apparent diffraction peak is observed for CNT-COOH in the XRD pattern of CNT-COO^−^/Ag_3_PO_4_@AgIO_4_. This could be due to their relatively low percentage in the composites.Alternatively, the functionalization process maynarrow those peaks due to the loss of amorphous carbon, which is in good agreement with the previous reports [23,37]. The XRD patterns confirmed the fabrication of the CNT-COO^−^/Ag_3_PO_4_@AgIO_4_ composite.

#### 2.2.3. SEM Analysis

The morphologies of the synthesized composites were analyzed viaSEM, and the results are shown in Figure 3. Appendix A of the Appendix A show images of CNT before and after purification, respectively. The carboxylated CNT-COOH was shown in Figure 3a, with different lengths ranging between 100 nm and 1µm. Figure 3b shows the image of the cubic-like morphology of Ag_3_PO_4_. AgIO_4_ displayed a hexagonal microstructure (Appendix A). In Figure 3c, it can be seen that the micro-size AgIO_4_ was successfully coupled with Ag_3_PO_4_ particles. Appendix A shows the SEM image of CNT/Ag_3_PO_4_@AgIO_4_. Figure 3d,e represented a different magnification of the SEM images of CNT-COO^−^/Ag_3_PO_4_@AgIO_4_. As seen, the Ag_3_PO_4_@AgIO_4_ particles successfully decorated the CNT-COO surface. Moreover, the particle size decreased compared tothe individual one. This is beneficial forenhanced photocatalytic activity. The EDS element spectra of the CNT-COO^−^/Ag_3_PO_4_@AgIO_4_ composite are shown in Figure 3f, which further confirmed the presence of the C, O, Ag, P, and I elements on the as-synthesized CNT-COO^−^/Ag_3_PO_4_@AgIO_4_.

#### 2.2.4. Optical Properties

The optical properties of the synthesized photocatalysts were evaluated using DRS analysis. The DRS spectra of Ag_3_PO_4_, Ag_3_PO_4_@AgIO_4,_and CNT-COO^−^/Ag_3_PO_4_@AgIO_4_ are shown in Figure 4a. The absorption edges of CNT-COO^−^/Ag_3_PO_4_@AgIO_4_, Ag_3_PO_4_@AgIO_4,_ and Ag_3_PO_4_ were observed at about 586, 455, and 543 nm, respectively. The light absorption of Ag_3_PO_4_@AgIO_4_ was extended in the visible light region compared with that of pure Ag_3_PO_4_. This is due to the combination of the two silver salts. The absorption edge of CNT-COO^−^/Ag_3_PO_4_@AgIO_4_ was further extended to 586 nm with the introduction of the CNT-COO. This confirms that the CNT-COO^−^/Ag_3_PO_4_@AgIO_4_ composite strongly enhanced the absorption of the visible light of Ag_3_PO_4_ due to the good interfacial interface between CNT-COO^−^ and Ag_3_PO_4_@AgIO_4_, which is beneficial to the enhancement of the photocatalytic activity. The band gaps of the synthesized composites are determined by plotting the transformed Kubelka–Munk function of light energy (αhʋ)^2^ versus energy (hʋ), (see Figure 4b). The band gaps were estimated to be 2.05 eV, 2.23 eV, 2.4 eV, and 2.45 eV attributed to CNT-COO^−^/Ag_3_PO_4_@AgIO_4_, Ag_3_PO_4_@AgIO_4,_ and Ag_3_PO_4_@AgIO_4_, respectively.

#### 2.2.5. Electrochemical Impedance Spectroscopy

The electron–hole recombination resistance and the charge transfer ability ofthe synthesized photocatalysts were measured using electrochemical impedance spectroscopy (EIS). As shown in Figure 5, the diameters of the arc followed the order CNT-COO^−^/Ag_3_PO_4_@AgIO_4_ < CNT-Ag_3_PO_4_@AgIO_4_ < Ag_3_PO_4_@AgIO_4_ < Ag_3_PO_4_. Therefore, it can be confirmed that CNT-COO^−^/Ag_3_PO_4_@AgIO_4_ exhibited enhanced e^−^/h^+^ separation and transferability at the catalyst–electrolyte interface compared with the other composites. In addition, the good interaction between the CNT-COO and Ag_3_PO_4_ in the CNT-COO/Ag_3_PO_4_@AgIO_4_ composite further enhanced the electron transfer over the CNT-Ag_3_PO_4_@AgIO_4_, which was due to the interfacial bonding between the CNT-COO and Ag_3_PO_4_, which resultedin the higher photocatalytic activity of CNT-COO^−^/Ag_3_PO_4_@AgIO_4_.

### 2.3. Photocatalytic Activity

#### 2.3.1. Photocatalytic Degradation Results and Analysis

The photocatalytic activities of the synthesized composites were evaluated based onthe photocatalytic degradation of MB under natural sunlight irradiation. To investigate the efficient catalytic activity between different CTN-based composites, their photodegradation performance toward MB was carried out under natural sunlight. The carboxylatedCNT-based composite (CNT-COO^−^/Ag_3_PO_4_@AgIO_4_) exhibited higher activity than the non-carboxylated one (CNT/Ag_3_PO_4_@AgIO_4_), Figure 6a.This is because the surface electric properties of CNT-COO greatly increased the dispersity of the catalyst in the aqueous dye solution (Appendix A), which increased the contact between the dye and the catalyst. Moreover, the adsorption of the positively charged dye on the CNT-COO^−^/Ag_3_PO_4_@AgIO_4_ increased due to the electrostatic attraction between the different charges (Appendix A). Additionally, the electrostatic interaction between CNT-COO and Ag_3_PO_4_ significantly enhanced the charge transfer, which therefore decreasedthe recombination between the photogenerated electron–hole pairs. This is in good agreement with the EIS results.

To investigate the optimal CNT-COOH amount on the synthesized composites, the photocatalyticperformanceofCNT-COO^−^/Ag_3_PO_4_@AgIO_4_-2.5%, CNT-COO^−^/Ag_3_PO_4_@AgIO_4_-5%, and CNT-COO^−^/Ag_3_PO_4_@AgIO_4_-7.5% has been tested in relation tothe degradation of MB (Appendix A). The results revealed that CNT-COO^−^/Ag_3_PO_4_@AgIO_4_-5% exhibited the highest catalytic activity. When the content of CNT-COO increased to 7.5 mg, the degradation efficiency wasreduced. This may be due to the increasing amount of CNT-COO^−^ shielding the surface of the photosensitive Ag_3_PO_4_ from the light, and the dye may have beenisolated from direct contact with the catalyst surface.

A photocatalytic activity comparison study was carried out between CNT-COO^−^/Ag_3_PO_4_@AgIO_4_-5% and its contents, Ag_3_PO_4_@AgIO_4_ and Ag_3_PO_4_ to further prove the photocatalytic efficiency of CNT-COO^−^/Ag_3_PO4@AgIO_4_-5%. As seen in Figure 6b, in the absence of the photocatalyst and/or light irradiation, the dye degradation can be ignored. CNT-COO^−^/Ag_3_PO_4_@AgIO_4_ exhibited the highest photodegradation of the dye, with nearly 100% of MB decomposingwithin 4 min, while the Ag_3_PO_4_@AgIO_4_ and the bare Ag_3_PO_4_ displayed degradation efficiencies of 70% and 40% within 4 min, respectively. The enhanced photocatalytic activity is attributed to the excellent charge separation and transferring of CNT-COO^−^/Ag_3_PO_4_@AgIO_4_-5% composite.

#### 2.3.2. Simultaneous Degradation of Different Organic Dyes

The photocatalytic activity of the as-synthesized composite toward the decomposition of a mixture of organic dyes was investigated under natural sunlight. MO and Rh B with a concentration of 0.01 g/L are used as representative dyes. As clearly seen in Appendix A, nearly 90% of MO degraded within 2 min, while 60% of Rh B degraded at the same time. The results further confirm the enhanced catalytic activity of the synthesized composite.

#### 2.3.3. Photocatalytic Reaction Kinetics

The kinetic behavior of the synthesized composite on the degradation of the organic dyes under light irradiation could be expressed as follows:−ln(Ct/Co) = kt(1)
where k is the degradation rate constant, Co is the initial concentration of MB, and Ct is the concentration of the dye at the irradiation time of t. Figure 7 shows the regression curves of-ln(Ct/Co) versus irradiation time, indicating that the photodegradation of MB over different samples is considered to fit pseudo-first-order kinetics. The MB degradation rate constants under different conditions are shown in Table 1. As seen, CNT-COO^−^/Ag_3_PO_4_@AgIO_4_-5% composite has predominantly enhanced photocatalytic performance for the degradation of MB with a degradation rate constant of 0.877min^−1^ compared to CNT/Ag_3_PO_4_@AgIO_4_,Ag_3_PO_4_@AgIO_4_, and Ag_3_PO_4_, which have a rate constant of 0.4143 min^−1^ 0.3107 min^−1^ and 0.1611 min^−1^, respectively.

To study the effect of the light intensity on the photocatalytic activity of CNT-COO^−^/Ag_3_PO_4_@AgIO_4_-5%, a set of photocatalytic experiments were carried out using a 350 W Xe lamp as a light source. As seen in Appendix A, the photocatalytic activity of CNT-COO^−^/Ag_3_PO_4_@AgIO_4_-5%is increased with the increase of the light intensity, and the degradation rate constants of the dye are0.877 min^−1^, 0.794 min^−1^, 0.595 min^−1^ and 0.309 min^−1^, corresponding to the light intensities of 100%, 75%, 50%, and 25%, respectively.

#### 2.3.4. Catalyst Recycling

To investigate the reusability of the synthesized composite, the CNT-COO^−^/Ag_3_PO_4_@AgIO_4_-5%was repeatedly exposed to degraded MB solution. As mentioned in the photocatalytic activity study, CNT-COO^−^/Ag_3_PO_4_@AgIO_4_-5% was suspended in the dye solution and then subjected to natural sunlight illumination, and the degradation of the dye was determined as mentioned above. This process was repeated for three cycles. After each cycle, the composite was washed using deionized water and dried at 60 °C, then used for the next cycle. As seen in Figure 8, the synthesized catalyst displayed efficient reusability and stability during three successive cycles.

#### 2.3.5. Photocatalytic Activity Mechanism

To investigate the effect of reactive species (h^+^, O_2_^•−^, and ^•^OH) on the photodegradation of organic pollutants, reactive species trapping experiments were carried out.It should be noted that 1 mM of Na_2_-EDTA, BZQ, and tert-butanol were used for scavenging h^+^, O_2_^•−^, and ^•^OH, respectively. The experimental results in Figure 9 revealed that the addition of the scavengers to the reaction systemreduced the dye degradation efficiency, whereas, the introduction of BZQ and Na_2_-EDTA into the reaction system reduced the photocatalytic activity of the CNT-COO^−^/Ag_3_PO_4_@AgIO_4_-5% composite. Therefore, in this case, the photocatalytic degradation of the organic dyes over the CNT-COO^−^/Ag_3_PO_4_@AgIO_4_-5% composite mainly depends on the h+, the O_2_^•−^, and the ^•^OH.

#### 2.3.6. Possible Mechanism

The photocatalytic mechanisms of semiconductor composites can be illustrated by the Z-scheme theory and the heterojunction energy band theory [38]. Figure 10 shows the possible e^−^/h+ transfer pathways based on the two mechanisms. Under natural sunlight irradiation, both Ag_3_PO_4_ and AgIO_4_ could be excited to form the e^−^/h^+^pairs. According to the proposed heterojunction energy-band theory mechanism in Figure 10a, the electrons in the CB of Ag_3_PO_4_ can be transferred to the CB of AgIO_4_ easily, while the holes in the VB of AgIO_4_ can be transferred to the VB of Ag_3_PO_4_ due to the potentials of the CB and VB positions of AgIO_4_ (CB = 0.96 eV and VB = 3.61 eV) [18] are lower than those of the Ag_3_PO_4_ (CB = 0.45 eV and VB = 2.85 eV) [13,39].Therefore, the electrons concentrate on the CB of AgIO_4_. However, the CB of AgIO_4_ is more positive than the reduction of the oxygen E0 (O_2_/O_2_^•−^) (0.13 eV, vs. NHE) [40]. Therefore, the O_2_^•−^could not be generated. However, the scavenger study proved the significant effect of the O_2_^•−^on the dye degradation process, whichindicatesthat the photocatalytic activity mechanism of CNT-COO^−^/Ag_3_PO_4_@AgIO_4_ composite could not be illustrated by the heterojunction energy-band theory. The mechanism proposed by the Z-scheme theory described in Figure 10b can be obtained intwo ways. First, atthe contact interface between Ag_3_PO_4_ and AgIO_4_, many defects can be aggregated. Therefore, the quasi-continuous energy levels can be formed in the contact interface Ag_3_PO_4_–AgIO_4_. This led to the contact interface exhibiting the conductor properties, such as the formation of the Ohmiccontact [38]. Second, as observed in the SEM analysis, Ag_3_PO_4_ and AgIO_4_ have a matching surface edge. Thus, Ag^0^can be formed at the contact interface between Ag_3_PO_4_ and AgIO_4_. Therefore, during the photocatalytic reaction, the Ag_3_PO_4_–AgIO_4_ contact interface develops to be the recombination centerof the photoexcited electrons from CB of AgIO_4_ and the holes from VB of Ag_3_PO_4_. As a result, the photoexcited electrons on the CB of Ag_3_PO_4_can be easily transferred to the CNT, leading to enhanced charge separation. This is in good agreement with the EIS, DRS, and photocatalytic activity study results. Therefore, the Z-scheme theory provides more supportstothe photocatalytic mechanism pathway of our catalyst. The same illustration has also been proved in previous studies [41,42,43].

From the above results, it isclear thatthe presence of CNT-COO^−^ on the as-synthesized composite and the multi-electron–hole species generated by Ag_3_PO_4_ and AgIO_4_ played a significant role in the improvement of the photocatalytic activity of CNT-COO^−^/Ag_3_PO_4_@AgIO_4_. The main photocatalytic reaction for the photodegradation of the organic dyes can be described by the following equations:Photocatalyst + hʋhVB + eCB(2)
hVB^+^ + dyeCO_2_ + H_2_O^+^ intermediates(3)
hVB^+^ + H_2_O^•^OH + H^+^
(4)
O_2_ + e CB- O_2_^•^(5)
O_2_^−^ + dye CO_2_ + H_2_O^+^ intermediates(6)

As explained in Figure 10b, after the photocatalyst is exposed to sunlight, the CNT-COO^−^/Ag_3_PO_4_@AgIO_4_ composite generates multi-electron/holes (Equation (2)). AgIO_4_ excites electrons to its CB, leaving holes in the VB;these electrons recombine with the holes generated by Ag_3_PO_4_ in the contact interface between the two semiconductors. The photoexcited electron on the CB of Ag_3_PO_4_ further migrates to the functionalized CNT. As a result, the holes concentrate in the VB of AgIO_4_ and the electrons concentrate on the functionalized CNT. Then, the h^+^ can directly interact with the dye (Equation (3)), and may also decompose water molecules into ^•^OH radicals (Equation (4)). Meanwhile, the electrons on the surface of CNT can be scavenged by the dissolved O_2_ to form superoxide radicals (O_2_^•−^) (Equation (5)). Finally, these photogenerated species can effectively degrade MB into CO_2_, H_2_O, and other intermediates (Equation (6)) [44,45,46,47].

As discussed above, it can be concluded that the enhancement of the photocatalytic activity of the CNT-COO^−^/Ag_3_PO_4_@AgIO_4_ composite should be attributed to the significant coeffects between the Ag_3_PO_4_@AgIO_4_ nanoparticle and CNT-COO. First, an Ag_3_PO_4_ particle with a narrower band gap (2.05 eV) and exceptional absorption in the visible and near UV regions qualifies the composite with a high capability for harvesting light. In addition, the more efficient transfer of photogenerated electrons from excited Ag_3_PO_4_@AgIO_4_ to the dye molecules under light irradiation enhanced the photocatalytic efficiency.

Second, the interlayer between Ag_3_PO_4_@AgIO_4_ and CNT-COO can prevent the recombination of thephotoexcited electron–hole pairs. Third, the existence of H-bonds and the non-covalent intermolecular π–π conjugation between the MB and the CNT-COO^−^/Ag_3_PO_4_@AgIO_4_ composite can significantly improve the adsorption of the dye molecules and provide increasing interfacial contact. Therefore, offering more active adsorption sites and photocatalytic reaction centers is beneficial for the enhancement of photocatalytic performance [48,49].

## 3. Materials and Methods

### 3.1. Chemicals

AgNO_3_ was purchased from Tianjin Sailing Chemical Reagent Technology Co., Ltd., sodium hydrogen phosphate (dibasic and monobasic) was purchased from Aladdin Co., Ltd., disodium ethylene diamine tetra acetate (Na2-EDTA) was purchased from YantaishiShuangshuang Chemical Co., Ltd., tert-butanol was purchased from TianjinshiKaixin Chemical Co., Ltd., p-benzoquinone (BZQ) was purchased from TianjinshiGuangfu Chemical Reagent Institute, MB was purchased from TianjinshiTianxin Fine Chemical Industry Development Center, and HNO_3_ and H_2_SO_4_ were purchased from Sinopharm Chemical Reagent Co., Ltd. All the chemicals were used as purchased without further treatment, except for CNT, which was first purified by refluxing in nitric acid.

### 3.2. Preparation of Carboxylated CNT (CNT-COOH)

For the preparation of CNTs-COOH, CNT was first purified by refluxing in diluted nitric acid for 4 h. It was then sonicated in a mixture of 1:3 (*v*/*v*) HNO_3_ (70%)/H_2_SO_4_ (98%) for 8 h at room temperature. Afterwards, it was centrifuged at 11,000 rpm and dried at 70 °C for 12 h.

### 3.3. Synthesis of CNT-COO^−^/Ag_3_PO_4_@AgIO_4_

The CNT-COO^−^/Ag_3_PO_4_@AgIO_4_ composite was synthesized viathe room-temperature chemical fabrication method. Typically, carboxylated CNT (CNT-COOH) wassonicated in water for 30 min to obtain a water dispersion of CNT-COOH. Then, 1.2 g/20 mL of AgNO_3_ was added to the CNT desperation, which was stirred for 4 h in dark conditions. Afterward, 0.8 g/20 mLof Na_2_HPO_4_ was added drop by drop. After 1 h of reaction, a solution of KIO_4_ (0.04 g/20 mL) was added in drops, and the reaction contents were kept inthe same conditions for another 1h. Then, the precipitate was collected viafiltration and dried at 60 °C for 12 h.

### 3.4. Synthesis of CNT/Ag_3_PO_4_@AgIO_4_

For the synthesis of CNT/Ag_3_PO_4_@AgIO_4_composite, non-carboxylated CNT was sonicated in water for 2 h to obtain a water dispersion of CNT. Then, 1.2 g/20 mLof AgNO_3_ was added to the CNTs’ desperation under stirring for 4 h in dark conditions. Afterward, 0.8 g/20 mLof Na_2_HPO_4_ was added drop by drop. After 1 h of reaction, a solution of KIO_4_ was added in drops, and the reaction contents were kept at the same condition for another 1 h. Then, the precipitate was collected viafiltration and dried at 60 °C for 12 h. For comparison, Ag_3_PO_4_@AgIO_4_ and Ag_3_PO_4_ were synthesized following the same procedure without using CNT for the synthesis of Ag_3_PO_4_@AgIO_4_, and CNT and KIO_4_ for the synthesis of Ag_3_PO_4_.

### 3.5. Material Characterization

X-ray diffraction (XRD) measurements were carried out at room temperature using a Bruker D8-Advance X-ray powder diffractometer with Cu-Kα radiation (λ = 1.5406 A) in the 2θ range of 10 °C to 80 °C. The morphology and the composition were characterized using ascanning electron microscope (SEM) with an EDS system. The UV-visible diffuse reflectance (DRS) spectra were obtained on a Shimadzu UV-2450 spectrophotometer. The Fourier transform infrared (FT-IR) spectra of the samples were recorded on a Bruker Vertex 70 FT-IR spectrophotometer using the KBr method. The electron transfer properties of the synthesized composites were studied using an electrochemical impedance spectrometer (EIS) VMP2 multi-potentiostat with the Zsimpwin program.

### 3.6. Evaluation of the Photocatalytic Activity

The study of the photocatalytic activity of the synthesized photocatalysts was carried out using the MB solution at room temperature under natural sunlight. Briefly, 25 mg of the prepared photocatalyst was mixed with 50 mLof MB (0.01 g/L) andthen sonicated for 10 min to establish adsorption–desorption equilibrium. Afterward, the mixture was exposed to natural sunlight. During the illumination, 5 mLaliquots were collected at a given time interval andthen centrifuged (10,000 rpm, 10 min) to remove the photocatalyst. The concentrations of MB solutions were determined by the UV-visible spectrophotometer at 664 nm. Additionally, the effect of light intensity on the photocatalytic activity was tested using a 350 W Xe lamp as a light source.

The detection of the active species during the photocatalytic reaction was performed using p-benzoquinone (B.Q), disodium ethylene diamine tetra acetate (Na_2_-EDTA), and tert-butanol (t.B) as superoxide radical (^•^O_2_), holes (h^+^) and hydroxide radical (^•^OH) scavengers, respectively, followed by the photocatalytic activity test.

## 4. Conclusions

A novel CNT-COO^−^/Ag_3_PO_4_@AgIO_4_ was synthesized through electrostatic deposition of Ag_3_PO_4_ on the surface of the carboxylated carbon nanotube (CNT-COOH), followed by the growth of AgIO_4_. Compared with CNT/Ag_3_PO_4_@AgIO_4_, CNT-COO^−^/Ag_3_PO_4_@AgIO_4_ exhibited enhanced optical and charge transfer properties. The optimal CNT-COOH content on CNT-COO^−^/A_g3_PO_4_@AgIO_4_ was investigated; the composite with 5 mg of CNT-COOH exhibited the highest photocatalytic activity.Moreover, CNT-COO^−^/Ag_3_PO_4_@AgIO_4_ demonstrated enhanced photocatalytic performance compared to that of CNT/Ag_3_PO_4_@AgIO_4_, Ag_3_PO_4_@AgIO_4_, and Ag_3_PO_4_ under natural sunlight irradiation. The enhanced photocatalytic activity is attributed to the best light harvesting and inductionof electron–hole pair separation and transfer. The results of this study provide important information for the use of functionalized CNT-COOH in the field of photocatalysis. Furthermore, they present a new way to functionalize CNT usingdifferent functional groups, which may lead to further development in the field of photocatalysis. Moreover, this work could providea new way for using natural sunlight to facilitate the practical application of photocatalysts in environmental issues.

## Figures and Tables

**Figure 1 molecules-28-01586-f001:**
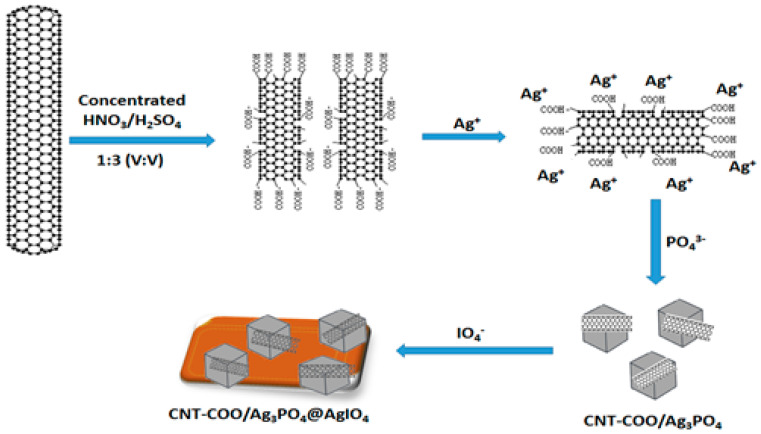
The scheme illustrates the synthesis method of the CNT-COO^−^/Ag_3_PO_4_@AgIO_4_ composite.

**Figure 2 molecules-28-01586-f002:**
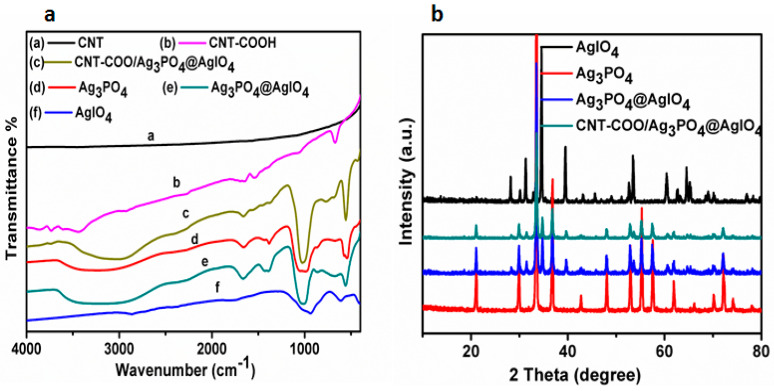
(**a**) The FT-IR spectra of the as-synthesized composites, (**b**) the XRD spectra of the as-synthesized.

**Figure 3 molecules-28-01586-f003:**
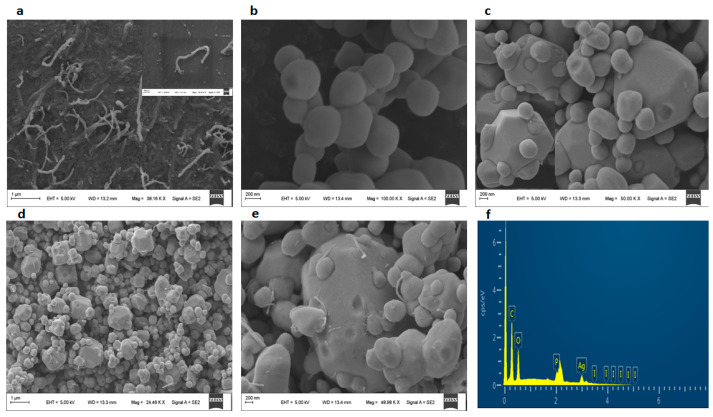
SEM image of (**a**) CNT-COOH, (**b**) Ag_3_PO_4_, (**c**) Ag_3_PO_4_@AgIO_4_, (**d**) CNT/Ag_3_PO_4_@AgIO_4_, (**e**) CNT-COO^−^/Ag_3_PO_4_@AgIO_4_, (**f**) EDS analysis of CNT-COO^−^/Ag_3_PO_4_@AgIO_4_.

**Figure 4 molecules-28-01586-f004:**
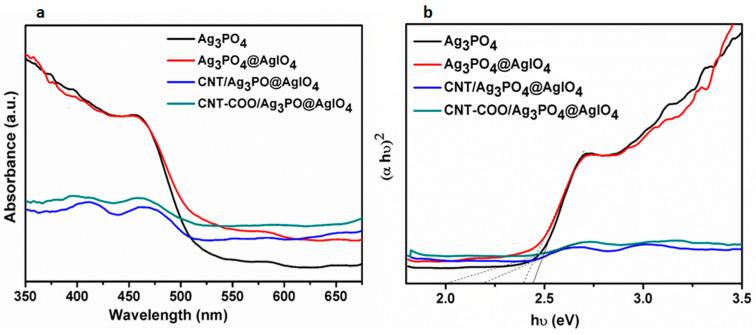
(**a**) DRS spectra of the synthesized materials, and (**b**) the plot of (αhʋ)^2^ versus energy (hʋ).

**Figure 5 molecules-28-01586-f005:**
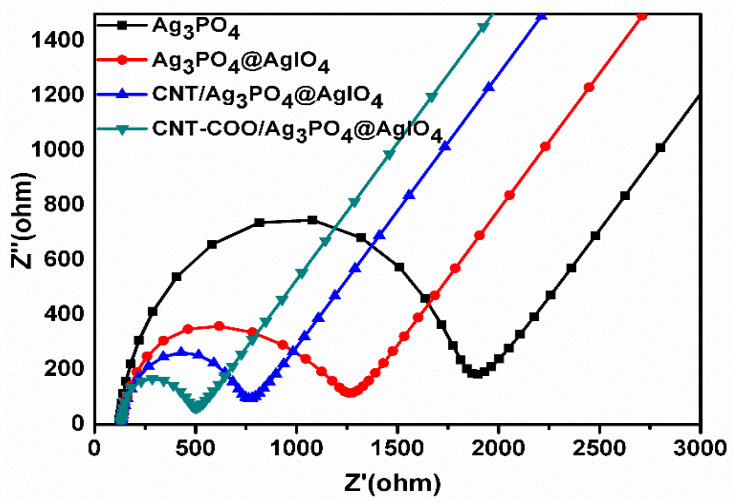
Impedance spectrum of the synthesized composites.

**Figure 6 molecules-28-01586-f006:**
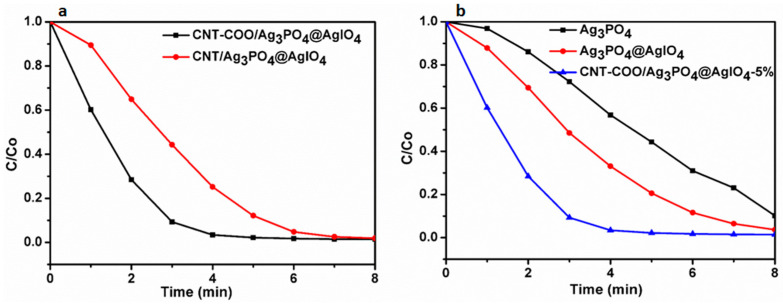
The photodegradation degradation of MB over (**a**) CNT-COO^−^/Ag_3_PO_4_@AgIO_4_ and CNT/Ag_3_PO_4_@AgIO_4_ (**b**) Ag_3_PO_4_, Ag_3_PO_4_@AgIO_4,_ and CNT-COO^−^/Ag_3_PO_4_@AgIO_4_, under sunlight irradiation.

**Figure 7 molecules-28-01586-f007:**
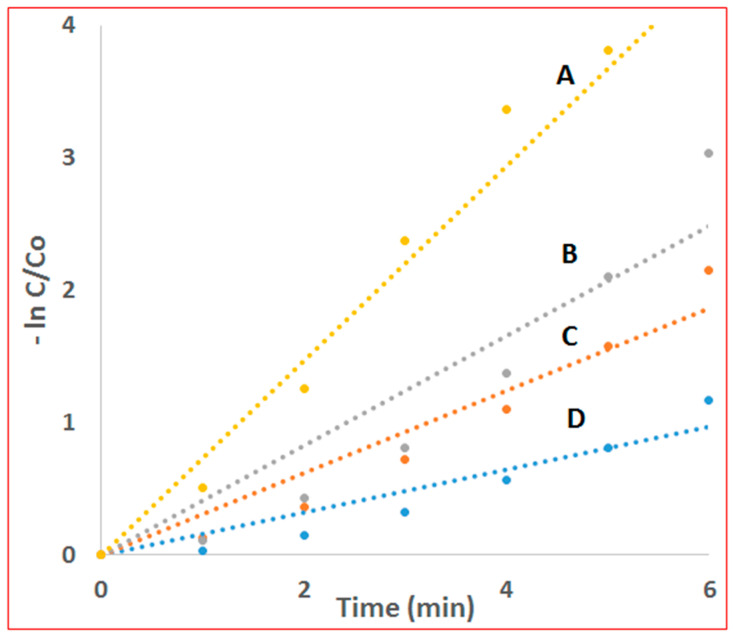
Regression curves of-ln(Ct/Co) versus irradiation time for (A) CNT-COO^−^/Ag_3_PO_4_@AgIO_4_-5%, (B) CNT/Ag_3_PO_4_@AgIO_4_, (C) Ag_3_PO_4_@AgIO_4_, (D) Ag_3_PO_4_.

**Figure 8 molecules-28-01586-f008:**
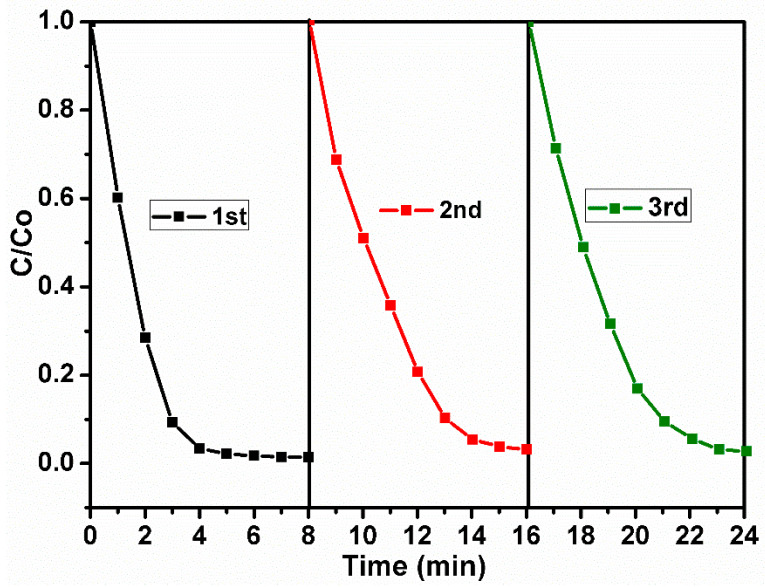
Stability and recycling of the CNT-COO^−^/Ag_3_PO_4_@AgIO_4_-5% composite.

**Figure 9 molecules-28-01586-f009:**
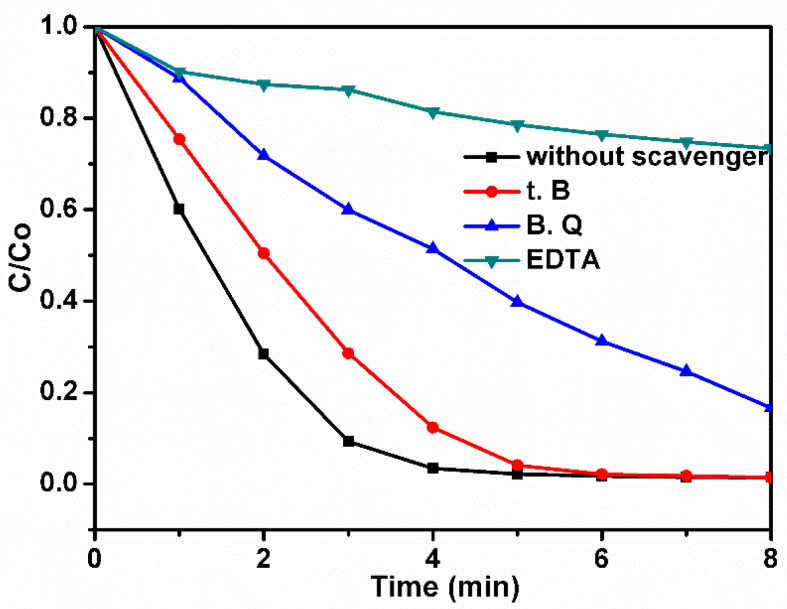
The effect of reactive species on the degradation of MB over CNT-COO^−^/Ag_3_PO_4_@AgIO_4_-5% composite under simulated light irradiation.

**Figure 10 molecules-28-01586-f010:**
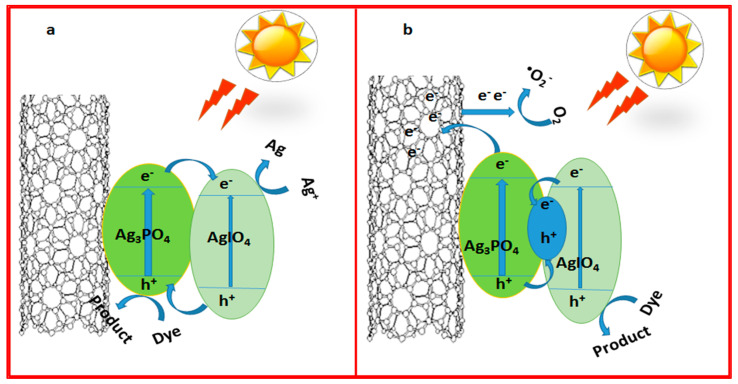
The proposed photocatalytic mechanism through which CNT-COO^−^/Ag_3_PO_4_@AgIO_4_ composite degrades organic dyes based on (**a**) heterojunction energy band theory and (**b**) Z-scheme theory under visible light irradiation.

**Table 1 molecules-28-01586-t001:** MB degradation rate constants over the synthesized composites.

Composite	Degradation Rate Constant (min^−1^)
CNT-COO^−^/Ag_3_PO_4_@AgIO_4_-5%	0.877
CNT/Ag_3_PO_4_@AgIO_4_	0.4143
Ag_3_PO_4_@AgIO_4_,	0.3107
Ag_3_PO_4_	0.1611

## Data Availability

All the data of this research were uploaded within the manuscript and in the Appendix A.

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
