# Peer review of "Fabrication of a Novel CNT-COO/Ag3PO4@AgIO4Composite with Enhanced Photocatalytic Activity under Natural Sunlight"

_molecules, 2023, doi:10.3390/molecules28041586_

Round 1

Reviewer 1 Report

The manuscript entitled “Fabrication of a novel CNT-COO/Ag3PO4@AgIO4 Composite with Enhanced Photocatalytic Activity under Natural Sunlightwritten by Elbashir et al., discusses how improved the photocatalytic activity when carboxylated carbon nanotube grafted Ag3PO4@AgIO4 was prepared through in situ electrostatic deposition method. The manuscript is divided into two parts.

In the first part, the authors present the analysis of x Fourier transform infrared (FT-IR) spectroscopy, X-ray diffraction (XRD), scanning electron 20 microscopy (SEM), and diffuse reflectance spectroscopy (DRS) techniques. Analyzing their data, the authors concluded the formation of the heterocomposite by FT-IR. The XRD patterns well confirmed the fabrication of CNT-COO/Ag3PO4@AgIO4 composite.  SEM images reveal the Ag3PO4@AgIO4 particles successfully decorated the CNT-COO surface. The EDS element spectra of CNT-COO/Ag3PO4@AgIO4 composite confirmed the presence of the C, O, Ag, P, and I elements. The band gap energy values were derived from DRS results suggesting that the CNT-COO/Ag3PO4@AgIO4 composite strongly enhanced the absorption of the visible light.

In the second part of the manuscript, the authors applied electrochemical impedance spectroscopy and evaluated the photocatalytic activity. The photocatalytic reaction kinetics can also be found, and an important discussion of the photocatalytic activity mechanisms.

The manuscript is generally scientifically sound. It is appropriately referenced and I find the level of detail as well as the choice of journal appropriate. I recommend the publication of this manuscript in molecules with major revisions.

I have some comments to improve the manuscript

In the abstract, the authors mention the x-ray photoelectron spectroscopy results. However, in the results and discussion section, these results were omitted. Include these results.

XRD section

Figure 2 panel b includes the Miller indices or includes the data related to JCPDS Card No. 06-0505 JCPDS Card No 10-0368.

Reference section

Ref 45 and 46 lines 546 and 547, respectively?

The authors did not consider in the manuscript superscripts and subscripts.

Author Response

Thank you for your valuable comments, please see the attachment.

Reviewer 2 Report

The following article is devoted to the fabrication of novel photocatalytic composite and its following testing in MB degradation. Regardless the area is important, the article is badly written, it contains numerous English and spelling mistakes, as well as mistakes in chemical formulas etc., which make the article unreadable. Before the submission in any journals it must be revised and significantly modified. Moreover, following comments can help improve the draft.
1.    The chemical formula of the composite presented as CNT-COO/Ag3PO4@AgIO4 is not clear as CNT-COO must be at least negatively charged or cation must be presented in the formula. 
2.    Writing of the formulas as Ag3PO4@AgIO4, Ag3PO4, PO43-, Ag+, cm-1 etc is not accessible, use subscript and superscript.
3.    Please revise the abstract; it contains numerous English and spelling mistakes.
4.    The Introduction is badly written, the importance of the investigation is not clear. The introduction contains numerous English and spelling mistakes. It is badly structured. No aims of the investigation is presented.
5.    Lines 77-79: “After the negatively charged CNT-COO was suspended in water and the Ag+ ions were added, the electrostatic interaction derived the adsorption of the positively charged Ag+ ions onto the negatively charged CNT-COO to form intermediate complexes of CNT-COO/Ag+.” How can you prove that all CNT-COOH reacted with Ag+ and there is no more –COOH groups in the system?
6.    Figure 1 is not clear. Add the description of the figure. CNT-COOH- does not have any sense and not exist in the Nature.
7.    Line 91. “Non-carboxylated CNT hasn’t any characteristic peaks” – English mistakes. Non-carboxylated CNT does not have any characteristic peaks
8.    Figure 2,a, add characteristic lines.
9.    Lines 309-313 are not clear and readable
10.  The mechanism of the reaction is not clear.

Author Response

Thank you, your comments were highly appreciated, they add value to the manuscript.

Reviewer 3 Report

In this paper, the authors report novel silver composite materials that can be applied as catalyst for photocatalytic reactions using sunlight as the light source. The work is well down, while the paper is well written. Therefore, it may be published after a minor revision:

(1)    The valence of the Ag element in the material should be studied by XPS

(2)    References are too old and need to be updated. The recent articles on silver catalysis should be involved: e.g. 10.6023/cjoc202011012

(3)    The mechanism is supposed to be a free radical mechanism. It should be proved by using free radical scavenger such as TEMPO. The details can be found in references,e. g. 10.1016/j.cclet.2020.09.012; 10.1016/j.cclet.2022.05.019. These references should be cited to support the mechanism.

(4)    The author supposed that hydroxyl radicals were involved. It should be confirmed by the capture experiment using salicylic acid. For details, please see: 10.6023/cjoc202008032 (Fig. 3c in the article)

Author Response

Thank you for your valuable comments

Round 2

Reviewer 1 Report

The authors took into account all the comments t improve the manuscript.

Author Response

Thank you for revising the revised version of the manuscript.

Reviewer 2 Report

The authors are modified the article. However, before the publishing, it must be significantly improved.

-Significant improvement of English is recommended. The article contains numerous English mistakes, incorrect sentences etc.

-The text of the article must be clearly revised once again. Several time, spaces between words are missed. For example, line 67.CNT-COO/Ag3PO4@AgIO4photocatalystwas, line 73 "CNT-COO-/Ag3PO4@AgIO4was", etc

-Line 66. "The photocatalyticphoto catalytic activity"

-Figure one contains COOH- (third picture), what means this negative charge on COOH?

-Line 49-50: "Due to their novel chemical, thermal, electrical and optical properties." Correct English

-Lines 103-114. No sense to add all characteristic lines in the text as it complicated reading. Please add them on the figure 2. Add also characteristic line on the IR spectra.

-Lines 310-314 are no readable.

Author Response

Thank you for revising the revised version of the manuscript.

please find the attached file: the response to the reviewer's comments.
